# PSSA: PCA-Domain Superpixelwise Singular Spectral Analysis for Unsupervised Hyperspectral Image Classification

Qiaoyuan Liu [1,†], Donglin Xue [1,†], Yanhui Tang [1], Yongxian Zhao [1,2], Jinchang Ren [3,4,*] and Haijiang Sun [1]

1 Changchun Institute of Optics, Precision Machinery and Physics, Chinese Academy of Sciences, Changchun 130033, China
2 University of Chinese Academy of Sciences, Changchun 130033, China
3 School of Computing Sciences, Guangdong Polytechnic Normal University, Guangzhou 510665, China
4 National Subsea Centre, Robert Gordon University, Aberdeen AB21 0BH, UK
* Correspondence: jinchang.ren@ieee.org
† These authors contributed equally to this work.

**Abstract:** Although supervised classification of hyperspectral images (HSI) has achieved success in remote sensing, its applications in real scenarios are often constrained, mainly due to the insufficiently available or lack of labelled data. As a result, unsupervised HSI classification based on data clustering is highly desired, yet it generally suffers from high computational cost and low classification accuracy, especially in large datasets. To tackle these challenges, a novel unsupervised spatial-spectral HSI classification method is proposed. By combining the entropy rate superpixel segmentation (ERS), superpixel-based principal component analysis (PCA), and PCA-domain 2D singular spectral analysis (SSA), both the efficacy and efficiency of feature extraction are improved, followed by the anchor-based graph clustering (AGC) for effective classification. Experiments on three publicly available and five self-collected aerial HSI datasets have fully demonstrated the efficacy of the proposed PCA-domain superpixelwise SSA (PSSA) method, with a gain of 15–20% in terms of the overall accuracy, in comparison to a few state-of-the-art methods. In addition, as an extra outcome, the HSI dataset we acquired is provided freely online.

**Keywords:** anchor-based graph clustering (AGC); hyperspectral image (HSI); singular spectral analysis (SSA); superpixels; unsupervised classification

## 1. Introduction

Hyperspectral image (HSI) classification has been successfully applied in a wide range of applications, such as military surveillance, forest survey, precision agriculture, food quality monitoring and industrial inspection [1–6]. With the fast development of portable devices, more and more breakthroughs have been made in emerging applications with unmanned aerial vehicles (UAV) and lab-based analysis [7–9]. However, their applications in real scenarios can be still quite limited, mainly due to the lack of sufficiently available labelled data for training and effective modelling and data classification, such as land mapping. Unsupervised learning is more suitable for remote sensing tasks with more unpredictable information, not limited to the hyperspectral image classification, but also for the multispectral and other hyperspectral remote sensing applications as well. With the advantage of not relying on any prior labeling information, the unsupervised approach can be more generic in a much wider range of applications while the time-consuming and costly process of data labelling can be avoided.

Currently most of the HSI classification approaches are based on supervised methods [10], where very good performance on standard datasets has been reported, e.g., 95–99% of the overall accuracy (OA) [11]. In contrast to these publicly available HSI data, in real scenarios the datasets are mostly unlabeled, where the pixel distribution tends to be more complex due to much overdetailed pixel information, unnecessary subclasses, shadows

and noise. This poses a great challenge to unsupervised HSI classification that cannot rely on any prior knowledge of labelled data.

Although unsupervised HSI classification is still in its infancy, it is clearly becoming a trend due to the relatively saturated performance of supervised methods [12–19]. Without any prior information as reference, existing approaches for unsupervised HSI classification are based mainly on three categories of data clustering techniques, i.e., the centroid-based clustering [20], the biological clustering [21] and the spectral clustering [22].

Centroid clustering includes K-means [23], fuzzy c-means [24], etc., and are relatively simple, classic and with strong operability, but are sensitive to initialization and noise, hence have an unsatisfactory classification accuracy in general. Biological clustering, such as artificial immune networks [25] and adaptive multi-objective differential evolution [26], mainly rely on the optimization strategy to realize classification accuracy improvement. However, their results may suffer from a high degree of randomness, caused by the random search in crossover and mutation operations, resulting in a big drawback of "poor reproducibility". Spectral clustering is more suitable for classifying data with arbitrary shapes, due mainly to the focus on the construction of an affinity matrix and eigenvalue decomposition [27,28], while achieving a high classification accuracy and robustness. As the size of the affinity matrix is decided by the number of samples in clustering, the eigenvalue decomposition could easily run out of computing memory when dealing with large-scale data such as HSI data. The most common method is to combine with the K-means method [29,30], so as to improve the efficiency by using a small number of principal components for clustering. However, the overall computational cost is still high while the performance is also sensitive to noise.

No matter the spatial or spectral domain, the redundant and noisy information contained in HSIs will worsen the difficulty of the unsupervised classification. To tackle these challenges, PCA-domain superpixelwise singular spectral analysis (PSSA) based unsupervised HSI classification is proposed. The major contributions of the proposed approach can be highlighted as follows.

(i) As a completely unsupervised approach, PSSA features several stages for improving the efficiency and efficacy of the extracted features, including decontaminating the data from over-segmented superpixels and jointly enhancing features in both the spatial and spectral domains based on superpixel-based PCA and PCA-domain 2D-SSA; it thus aims to preserve the discriminative spectral-spatial information while suppressing the noise and reducing the spectral redundancy and the data scale for more effective and efficient HSI classification.

(ii) To integrate the AGC to further improve the accuracy and efficiency of classification, with a gain of over 15% of the overall accuracy, as validated by experiments on several aerial remote sensing datasets, in comparison to a few state-of-the-art methods such as AGC [30] and NEC [31].

(iii) As an extra outcome, our self-collected HSI datasets are freely released online to benefit the community: https://zyx980824.github.io/HSI-dataset/ (accessed on 2 June 2021).

The rest of this paper is organized as follows. In Section 2, the theoretical background is presented, including the entropy rate superpixel (ERS), the 2D-SSA and the AGC. In Section 3, the proposed algorithm is discussed in detail. The datasets and experimental settings are given in Section 4, followed by the results and analysis in Section 5. Finally, some concluding remarks are drawn in Section 6 along with future directions.

## 2. Related Works

### 2.1. Notation

In this part, we define some notation to make sure that the mathematical meaning of the proposed method can be formulated clearly. The detailed notations are summarized in Table 1 and we explain the meaning of each term when it is first used.

**Table 1.** List of symbols used in the paper.

| | |
|---|---|
| $\mathcal{G}$ | weighted undirected graph |
| $V$ | Vertices |
| $E$ | Edges |
| $A$ | Subset of edges |
| $A^*$ | Binary map |
| $G$ | Trajectory matrix |
| X | Pixels |
| $\lambda$ | Eigenvalues |
| $U$ | Eigenvectors |
| $Y$ | Cluster indicator matrix |
| $y$ | Cluster indicator |
| Z | Similarity matrix |
| $L$ | Laplacian matrix |
| $I$ | Hyperspectral image |
| $s$ | Superpixel |

*2.2. Superpixel Image Segmentation*

Although superpixel-based image segmentation has been studied intensively, there are still limitations when applying them for unsupervised HSI classification. Based on the gradient descent theory, simple linear iterative clustering (SLIC) [32] is a typical superpixel segmentation algorithm that extracts superpixels according to the color similarity and the spatial distance of pixels. Although SLIC is simple and fast, it is overly dependent on the Red (R), Green (G) and Blue (B) or CIELAB color space in clustering pixels, often resulting in low edge consistency and poor noise robustness. Superpixels extracted from the energy-driven sampling (SEEDS) [33] is based on a regular image grid, where the results are gradually refined closely to the superpixel edges. Although SEEDS has potentially good edge consistency and noise robustness, its compactness is uncontrollable, leading to quite irregularly shaped superpixels. As another representative superpixel algorithm, linear spectral clustering (LSC) [34] uses a kernel function to measure the color similarity and spatial proximity for image segmentation. The texture edge of the generated superpixels can be well preserved, yet the shape of the segmented superpixels can also suffer from irregularity.

After comprehensive experimental comparison, the popular entropy rate superpixel segmentation (ERS) [35] algorithm was selected in our approach. For the ERS algorithm, a graph $G = (V, E)$ was constructed, whose vertices $V$ were the image pixels to be segmented, and the edge set $E$ consisted of the pairwise similarities by the weight function. The graph was partitioned into a connected subgraph by choosing a subset of edges $A \subseteq E$, such that the resulting graph $G' = (V, A)$ consisted of smaller connected components/subgraphs. The objective function of ERS is given below:

$$A^* = \underset{A}{\operatorname{argmax}} Tr\{H(A) + \alpha B(A)\}, \; s.t. A \subseteq E \tag{1}$$

where $Tr$ denotes the trace of the matrix, $H(.)$ is the entropy rate term of the random walk on the constructed graph as a criterion to obtain compact and homogeneous clusters, i.e., segmenting images on perceptual boundaries and favoring superpixels overlapping with only a single object; $B(A)$ is the balancing term defined as a monotonically increasing and submodular function on the cluster distribution to have similar sizes, i.e., reducing the number of unbalanced superpixels, and $\alpha \geq 0$ is the weight of the balancing term.

A greedy search algorithm can effectively solve the problem of maximizing the submodular objective function, finally achieving a reduced under-segmentation error up to 50%, a reduced boundary missing rate up to 40%, and a tighter segmentation accuracy bound [11]. In addition, ERS is highly efficient, taking ~2.5 s to segment an image of $481 \times 321$ pixels.

### 2.3. Brief of the 2D-SSA

The 2-D singular spectrum analysis (2D-SSA) has already shown its great potential in supervised HSI classification [16,36–38], which can help to successfully remove noise and thus, enhance the signal to noise ratio for improved accuracy. Given a band image with a size of h × w, let the embedding window have a size of u × v (with $1 \leq u \leq h$ and $1 \leq v \leq w$), which moves from the top left to the bottom right of the image to construct a trajectory matrix G. The pixels in the window are expanded and joined as a column vector $X \in R^{uv \times 1}$ in the trajectory matrix, as shown below:

$$G = (X_{1,1}, X_{1,2}, \ldots, X_{1,w-v+1}, X_{2,1}, \ldots, X_{h-u+1,w-v+1}) \in R^{uv \times (h-u+1)(w-v+1)} \quad (2)$$

Note that G has a structure called HbH, i.e., Hankel by Hankel.

The eigenvalues of $GG^T$ and the corresponding eigenvectors are denoted as $(\lambda_1 \geq \lambda_2 \geq \ldots \geq \lambda_L)$ and $(U_1, U_2, \ldots, U_L)$, respectively. The trajectory matrix can be written as:

$$G = G_1 + G_2 + \ldots + G_L$$
$$G_i = \sqrt{\lambda_i} U_i X_i^T, \quad V_i = G^T U_i / \sqrt{\lambda_i} \quad (3)$$

where $U_i$ and $X_i$ are the empirical orthogonal functions and the principal components of $G$, respectively.

The 2D-SSA can extract different components of the input image, including its trend $G_1$, oscillations, and noise, leading to smoothed and denoised data. Here, we select $G_1$ as an approximation to $G$, mainly because it contains the most important spatial information that benefits classification. Finally, $G_1$ is converted to a constructed new image of size $h \times w$ again, by a two-step diagonal averaging process [36].

By simultaneously considering both local and global spatial information, the 2D-SSA can effectively reduce the influence of image noise, and its reconstructed images have shown great noise robustness in HSI and non-HSI images [36,39].

### 2.4. The Anchor-Based Graph Clustering (AGC)

Inspired by the spectral clustering, the anchor-based graph clustering (AGC) is a representative approach for unsupervised HSI classification [30]. Similar to ERS, the input HSI is first represented as an undirected graph, in which each pixel is treated as a vertex, and their pairwise similarities are treated as the edges. The AGC realizes the HSI classification via multi-group spectral clustering by minimizing an objective function below:

$$\min \text{Tr}\left(Y^T L Y\right) + \lambda \parallel Y^T Y - I \parallel_F^2 \quad (4)$$

where $Y = [y_1^T, y_2^T, \ldots, y_n^T]$, and $\{y_1, y_2, \ldots y_c\}$ are the indicator vectors of pixels $\{x_1, x_2, \ldots x_c\}$; $\lambda > 0$ is the Lagrangian multiplier; $\parallel Y^T Y - I \parallel_F^2$ is the item for orthonormal constraint.; $L$ is the Laplacian matrix; $\text{Tr}(\cdot)$ is again the trace of the matrix; and F denotes the Frobenius norm.

As Equation (4) is a non-smooth objective function, an anchor-based graph is introduced. A label prediction function is set for the subset of anchor samples, which are obtained by a certain sampling interval s. Details about how s is decided are discussed in Section 4.3. The label prediction function f(.) with the subset $A = \{a_j\}_{j=1}^m$ is given below, and $a_j$ is the anchor sample.

$$f(x_i) = \sum_{j=1}^m Z_{ij} f(a_j) \quad (5)$$

where $Z$ is a similarity matrix between the whole samples and the chosen anchors. By relaxing the discreteness condition and considering the nonnegative and orthonormal constraints, Equation (4) can be rewritten as

$$LY + 2\lambda YY^T Y - 2\lambda Y = Q - P = \left(Y + 2\lambda YY^T Y\right) - \left(2\lambda Y + Z\Delta^{-1} Z^T Y\right) \quad (6)$$

where $Q = Y + 2\lambda YY^TY$ and $P = 2\lambda Y + Z\Delta^{-1}Z^TY$.

According to the standard nonnegative matrix factorization (NMF) [40], the minimization of the objective function in Equation (4) can be obtained by updating Y as follows:

$$Y \leftarrow Y \bigcirc P/Q \tag{7}$$

where $\bigcirc$ and P/Q denote, respectively, the Hadamard product and the Hadamard division (i.e., element-wise multiplication and division), and $Y_{ij} \leftarrow Y_{ij} \cdot P_{ij}/Q_{ij}$. We can obtain an optimal resolution until the objective function Equation (4) converges. As only one element is positive and all others approximate zero in each row of the indicator matrix $Y$, it can be considered as a nearly perfect indicator matrix for clustering representation.

## 3. The Proposed Algorithm

Figure 1 shows the flowchart of the proposed PSSA approach, which has three main key modules, i.e., superpixel segmentation guided data decontamination, PCA-domain 2D-SSA spectral-special feature enhancement, and AGC-based unsupervised HSI classification, as detailed below.

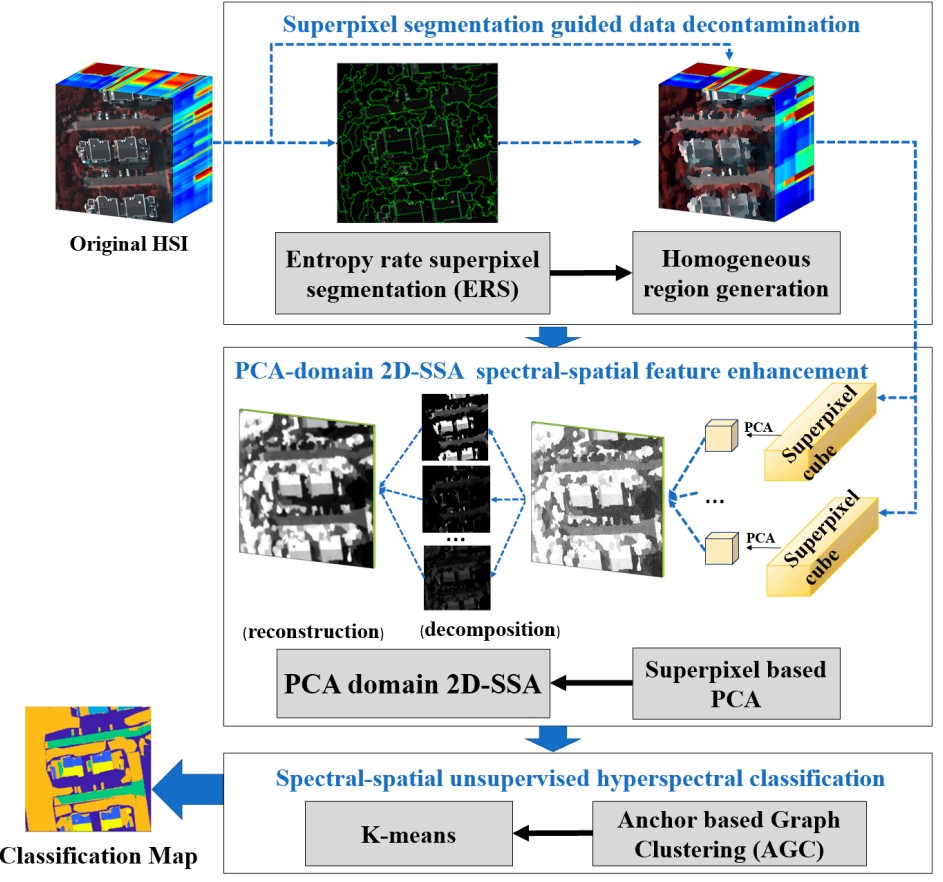

**Figure 1.** Outline of the PSSA framework for superpixelwise unsupervised HSI classification.

### 3.1. Superpixel Segmentation Guided Data Decontamination

In the process of labeling the ground truth, for convenience and ease of classification, neighboring pixels are often put into the regions of the same class. During supervised classification, the ground truth can be referred to for training a classifier. However, for unsupervised classification, due to the lack of ground truth, similarities between the neighboring pixels need to be considered. In HSI, superpixel segmentation is particularly useful to tackle noise and spectral inconsistency among neighboring pixels, even within the same class.

This superpixel-based pre-processing strategy, as shown in the block of data decontamination in Figure 1, can convert the input HSI into several homogeneous regions, using the ERS-based superpixel segmentation. The proposed strategy can not only remove the redundant information, but also help to merge fragmented patches into homogeneous regions, which is conductive to the subsequent classification. By comparing the experimental performance of various superpixel segmentation methods, the ERS method was selected as the most suitable one for our approach to extract homogeneous regions.

Let the size of the original HSI data $I_{in}$ be $M * N * B$, where $M$, $N$ and $B$ denote the width, height and number of bands of the HSI, respectively. Before the superpixel segmentation, the averaged band image is obtained to extract the dominant information $I_f$, of size $M * N$ [41]. Based on $I_f$, a segmentation map Y can be obtained as a binary image by ERS based segmentation, where 1 denotes the edges of the superpixels, and 0 denotes non-edge pixels.

By applying $Y$ to $I_{in}$, the input HSI can be segmented into superpixels as:

$$I_{in} * Y = \left\{ \bigcup_i^n s_i, s.t. \ s_i \cap s_j = \varnothing, \ (i \neq j), s_i \in I_{in} \right\} \tag{8}$$

where $s_i$ is the $i$th superpixel and $n$ is the number of superpixels.

To reduce the computational burden caused by the complex pixel distribution, several homogeneous regions are generated based on the over-segmented superpixel patches. In detail, the mean spectra of all pixels within the superpixel patch $s_i$ is calculated to replace the original pixels; therefore, the original HSI can be transferred to $I_1$ as follows:

$$I_1 = \left\{ \bigcup_i^n s_i', s.t. \ s_i' \cap s_j' = \varnothing, \ (i \neq j) \right\} \tag{9}$$

$$s_i' = mean(s_i), s_i \in I_{in} \tag{10}$$

The elements obtained in $I_1$ are actually the smoothed homogeneous region. In this way, the subsequent classification method could classify the more integrated data with less noise and intra-inconsistency, resulting in a reduction on the computational complexity and computational burden.

*3.2. PCA-Domain 2D-SSA Spectral-Spatial Feature Enhancement*

In order to further enhance the separability of HSI and the adaptability of the unsupervised classification to various volumes of data, the 2D-SSA is introduced for the first time for unsupervised HSI classification. During the process of generating homogeneous regions, the pixel values in each superpixel are set to the same average value of that patch. This can actually significantly save computational time, as we can use the superpixel-based PCA to reduce the dimension of the spectral data, rather than using each individual pixels while suppressing the noise and intra-class inconsistency.

With the extracted PCA components, a PCA-domain 2D-SSA is applied for spatial-spectral feature enhancement, where the dimension of the spectral data has been much reduced for efficiency. This can mitigate the common out-of-memory problem that exists in the current unsupervised classification methods when dealing with large-volume data. On the contrary, our strategy can successfully solve this issue.

In the proposed feature enhancement strategy, the superpixel-based PCA was implemented on only one pixel, i.e., the average one, for each superpixel, rather than all pixels within that patch. After spectral domain PCA, the number of bands was reduced from $B$ to $B'$, where $B' \ll B$ ($B'$ is set to 15 and $B$ is hundreds). By combining the dimension-reduced superpixel patches together, a new image $I'$ was obtained as

$$I' = \left\{ \bigcup_i^n s_i'', s.t. s_i'' = PCA\left( s_i', B' \right) \right\} \tag{11}$$

For each band within $I'$, a PCA-domain 2D-SSA was applied to further enhance the features and remove the noise. The refined HSI $I_2$ will be used for HSI classification.

$$I_2 = 2\text{DSSA}\left(I'\right) \tag{12}$$

*3.3. Spectral-Spatial Unsupervised HSI Classification*

Among existing unsupervised HSI classification methods, spectral clustering-based ones often show good performance, where the recently proposed anchor-based graph clustering (AGC) algorithm [30] is one of the most represented ones. For better segmentation accuracy and efficiency, we tested the two proposed strategies in the AGC framework. The AGC algorithm was also set as the baseline for comparison. Here, we briefly describe the implementation of the proposed PSSA with both the spatial and spectral redundant information eliminated, as summarized in Algorithm 1 below.

---

**Algorithm 1: Proposed unsupervised HSI classification**

---

**Input:** Hyperspectral image datasets $I_{in}$.
**Output:** Indicator matrices $Y$ and clustering result $S$.
Decontaminate $I_{in}$ to $I_1$ according to Equation (9);
Input $I_1$ to data optimization with 2D-SSA for $I_2$ according Equations (11) and (12)
Choose $m$ samples in $I_2$ for AGC construction
Calculate the matrix $Z$ according to Ref. [27]
**while** iterations $<= i$ **do**
Update $P$ and $Q$ with $Z$ by $P = 2\lambda Y + Z\Delta^{-1}Z^TY$
$\qquad$ and $Q = Y + 2\lambda YY^TY$
Update the indicator matrix $Y$ according to Equation (7)
**End**
Input $Y$ to the $K$-means to get the clustering result $S$.

---

The two strategies proposed were for preprocessing before classification. For the input HSI data $I_{in}$, the data decontamination strategy firstly transformed it into $I_1$, followed by the feature enhancement strategy that gained a new image $I_2$, which fed into the AGC framework for classification to achieve the final results. Eventually, we performed a comprehensive evaluation, though these simple and effective strategies are also suitable for other (unsupervised) classification frameworks.

## 4. Datasets and Experimental Settings

*4.1. Introduction to the Datasets*

We conducted experiments on three publicly available HSI datasets, including Salinas, Salinas-A, and Indian Pines, as well as five self-collected datasets, covering various land-cover classes at different spatial and spectral resolutions.

The Salinas dataset was acquired by the 224-band AVIRIS sensor over the Salinas Valley, California, and is characterized by a high spatial resolution (3.7-m pixels). We used the 204 bands, after eliminating the water absorption and noisy bands (108–112, 154–167 and 224). Salinas has $512 * 217$ pixels, and the ground truth contains 16 classes. SalinasA is a subset of the Salinas, having only $86 \times 86$ pixels in six classes [42,43].

The Indian Pines dataset was also collected by the AVIRIS sensor [44] in Northwestern Indiana and consists of $145 \times 145$ pixels and 224 spectral reflectance bands. It contains two-thirds agriculture, and one-third forest or other natural perennial vegetation. The ground truth has 16 classes, and only 200 bands are used, after removing the noisy and water absorption ones, along with a quite low spatial resolution of 20 m [45].

The self-collected HSI datasets, shown in Figure 2, were gathered by the Gaiasky mini2 VN HSI camera; each of them contains $500 \times 500$ pixels and 360 spectral reflectance bands. These are actually a cropped subscene from a large image of $3091 \times 3129$ pixels, collected at the gate of the Shanghai Drama Academy in Qingdao in June 2020, with a very high

spatial resolution of 16 cm. The spectral range is 393–1011 nm with a spectral resolution of 3.5 nm, consisting of observations from 10 identified classes, i.e., building, tree, road, car, playground, grass, brick house, iron house, steel sports hall, and land.

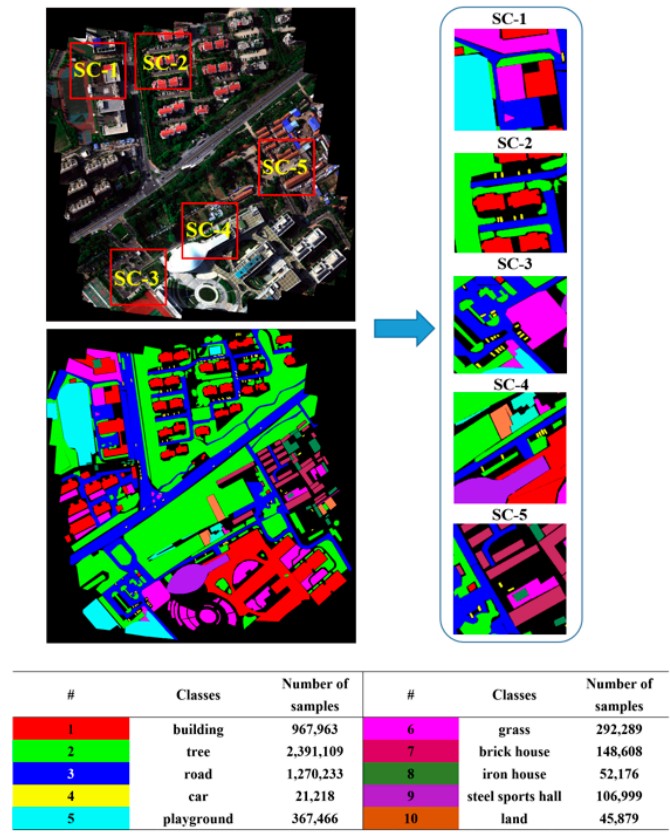

| # | Classes | Number of samples | # | Classes | Number of samples |
|---|---------|-------------------|---|---------|-------------------|
| 1 | building | 967,963 | 6 | grass | 292,289 |
| 2 | tree | 2,391,109 | 7 | brick house | 148,608 |
| 3 | road | 1,270,233 | 8 | iron house | 52,176 |
| 4 | car | 21,218 | 9 | steel sports hall | 106,999 |
| 5 | playground | 367,466 | 10 | land | 45,879 |

**Figure 2.** Examples of the self-collected HSI datasets, where SC-1 to SC-5 are small patches of $500 \times 500$ pixels each, cropped from a large hypercube of $3091 \times 3129$ pixels.

### 4.2. Evaluation Metrics

In our experiments, five widely used evaluation metrics were used for quantitative assessment, including the Purity (P.), the Normalized Mutual Information (NMI), overall accuracy (OA), average accuracy (AA), and the Kappa coefficient, as briefly described below.

The P. is the most common metric for evaluation of clustering results, given by:

$$\text{Purity}(\Omega, \hat{\Omega}) = \frac{1}{n} \sum_j \max_j |\Omega_i \cap \hat{\Omega}_j| \tag{13}$$

where $n$ is the number of clustering sets, $\Omega$ is the clustering result set and $\hat{\Omega}$ is the ground truth, and $i$ and $j$ are the indices of the clustering set. The value lies within [0, 1] and the higher the better.

The NMI score is defined as follows:

$$\text{NMI} = \frac{\sum_{i=1}^{c} \sum_{j=1}^{c} n_{i,j} log \frac{n_{i,j}}{n_i \hat{n}_j}}{\sqrt{\left(\sum_{i=1}^{c} n_i log \frac{n_i}{n}\right)\left(\sum_{i=1}^{c} \hat{n}_i log \frac{\hat{n}_i}{n}\right)}} \tag{14}$$

where $n_i$ denotes the number of data contained in the cluster $C_i (1 \le i \le c)$, $\hat{n}_j$ is the number of data belonging to the $L_j (1 \le j \le c)$, and $n_{i,j}$ denotes the number of data that are in the intersection between the cluster $C_i$ and the class $L_j$. The larger the NMI, the better the clustering result.

The overall accuracy $(OA)$ is the percentage of all pixels that are correctly classified, and the average accuracy $(AA)$ stands for the average percentage of correctly classified pixels for each class. Considering that in $C$ classes the number of correctly classified pixels for each class is $N_i$ and the total number of pixels for each class is $T_i$, we can have

$$OA = \frac{\sum_{i=1}^{C} N_i}{\sum_{i=1}^{C} T_i} \tag{15}$$

$$AA = \frac{1}{C} \sum_{i=1}^{C} \frac{N_i}{T_i} \tag{16}$$

The Kappa coefficient provides a standard for the overall classification performance by comparing the agreement against the one occurring by chance, which is defined by

$$kappa = \frac{OA - p_e}{1 - p_e} \tag{17}$$

where $p_e$ is the proportion agreement occurring by chance. Assuming the real labels in each class are $a_1, a_2, \ldots a_C$, with the predicted labels being $b_1, b_2, \ldots b_C$, the total number of pixels is $n$, then $p_e$ is defined as:

$$p_e = \frac{a_1 \times b_1 + a_2 \times b_2 + \ldots + a_C \times b_c}{n \times n} \tag{18}$$

The value of Kappa ranges from $-1$ to 1, where $-1$ stands for complete disagreement and 1 stands for perfect agreement.

*4.3. Implementation Details*

The PSSA was implemented using the Matlab2019b platform, on a PC with 1.70 GHz Intel(R) Core(TM) i5-8350U without a GPU. There were three preset key parameters in our method, which included (i) the number of superpixels n in ERS (Equation (8)), (ii) the sampling interval s (Section 2.3), and (iii) the number of iterations $i$ in Algorithm 1. The performance of our method was evaluated in various combinations, including the baseline (B), baseline+data decontamination (B + DD), baseline+data optimization (B + DO), and the full algorithm (B + DO + DD).

Figure 3 shows the corresponding classification results with different numbers of superpixels in terms of NMI and P. As seen, adding one of the proposed strategies alone achieved a certain degree of improvement over the baseline, yet the results can be much improved when applying both strategies. In addition, the most appropriate number of superpixels was related to the size of the data. For Salinas with a spatial size of $512 * 217$, its most appropriate number of superpixel was 500. However, for SalinasA with $86 * 86$ pixels and India Pines with $145 * 145$ pixels, the recommended number of superpixel becomes 100. In contrast, both too coarse or too detailed segmentation will lead to degraded performance after data smoothing, and thus affect the subsequent classification.

Figure 4 illustrates the NMI and P. of the proposed algorithm with different anchor sampling interval n, whose value is chosen from {30, 40, 50, 60}. As seen, based on the improvement achieved step-by-step, the smaller the sampling interval, the better the classification results obtained. However, a smaller sampling interval will increase the scale of processed data and delay the processing speed. As a result, 40 was chosen as the sampling interval parameter.

Figure 5 illustrates the NMI and P. of the proposed algorithm at different numbers of iterations i, whose value varies from 0 to 200 with an interval of 10. It can be concluded that the number of iterations has a limited impact on the classification results, but combining the two strategies together can produce a more robust classification than using one strategy alone.

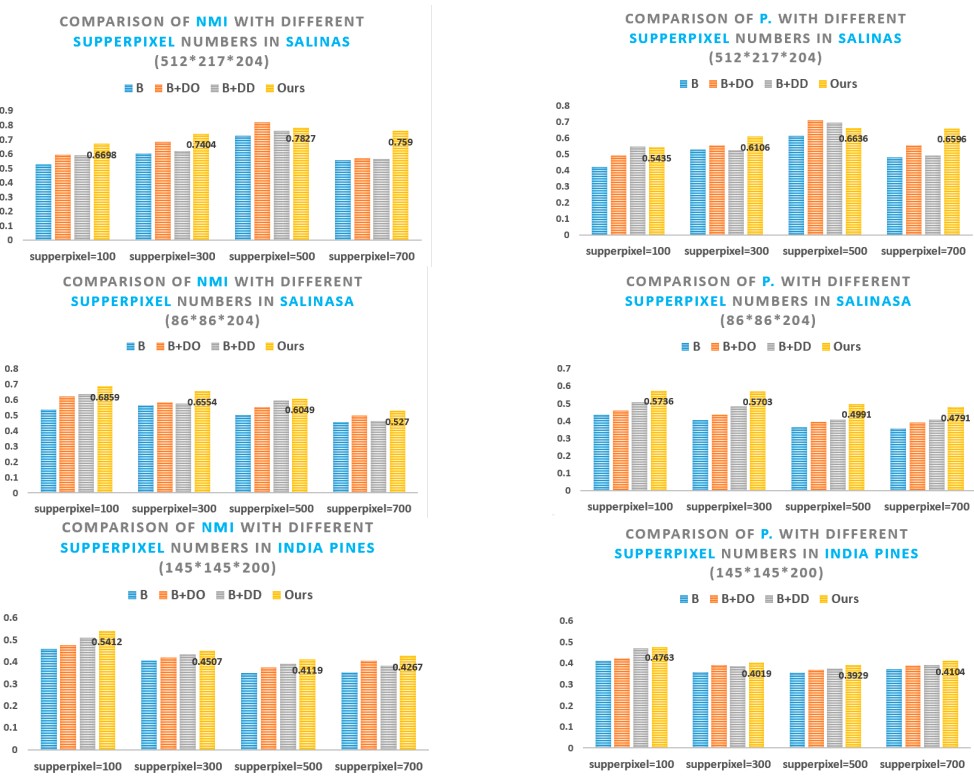

**Figure 3.** Classification results with different superpixels.

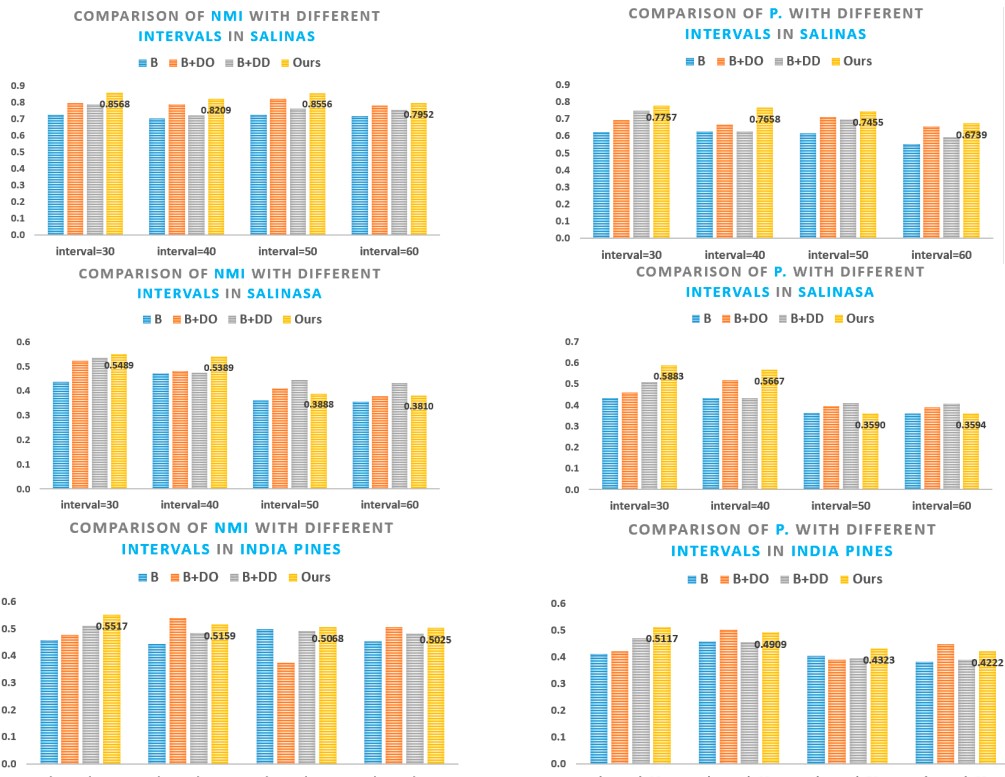

**Figure 4.** Classification results in different sampling interval.

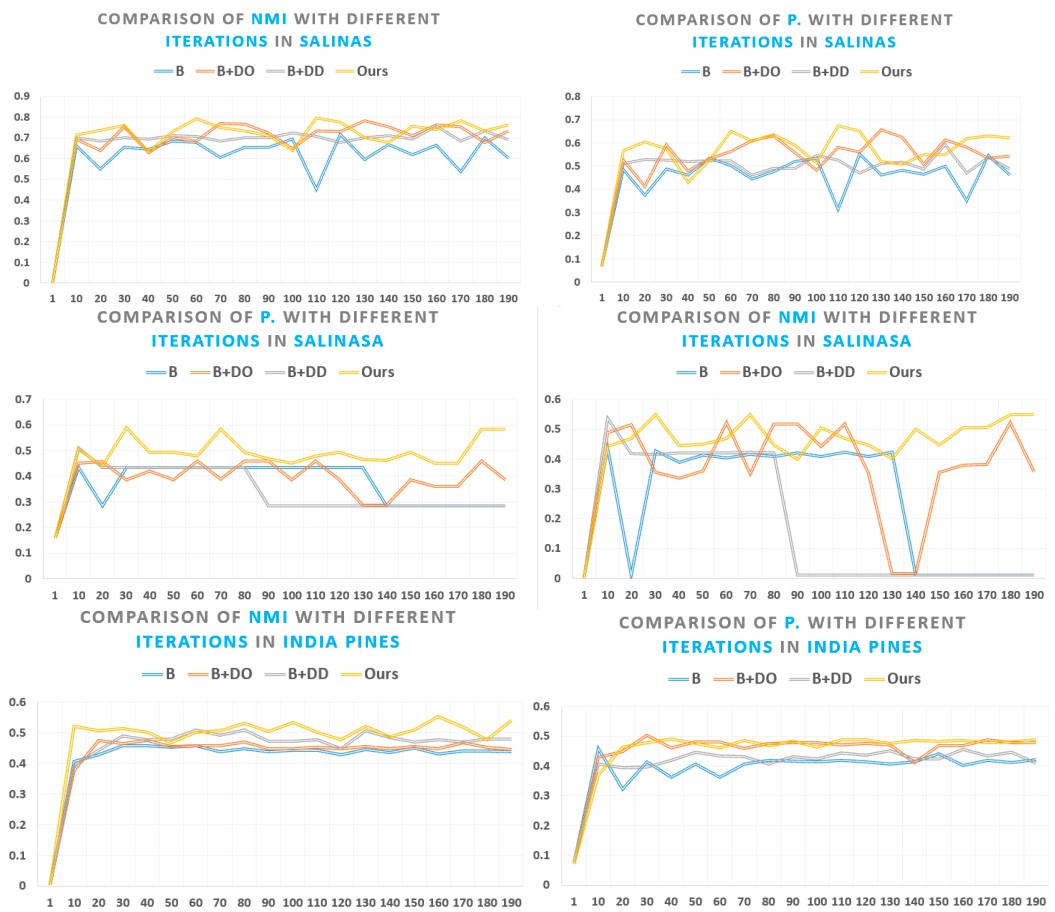

**Figure 5.** Classification results in different numbers of iterations.

## 5. Experimental Results and Discussion

### 5.1. Superpixel Segmentation Analysis

In this section, a further study is presented to show the merit of the adopted ERS superpixel method, in comparison to three commonly used superpixel methods, i.e., SLIC [32], Seeds [33] and LSC [34]. Specifically, we compared our approach on the Salinas, SalinasA and India Pines datasets, and the results are given in Figure 6. As seen, ERS in general outperformed others as it could handle different land-cover classes in HSI more consistently and compactly, benefiting the final classification.

In addition to the intuitive comparison, we also performed a quantitative evaluation using the final classification results to further illustrate the efficacy of the ERS in Table 2. As seen, the ERS-based unsupervised HSI classification achieved the highest NMI and P. values on both the public and the self-collected HSI datasets, making it more suitable than the other three in superpixel-based unsupervised HSI classification.

**Table 2.** Results from various superpixel methods, results in bold highlight the best in the group.

|  | Indian Pines | | SalinasA | | Salinas | |
|---|---|---|---|---|---|---|
|  | **NMI** | **P** | **NMI** | **P** | **NMI** | **P** |
| LSC | 0.574 | 0.497 | 0.773 | 0.705 | 0.691 | 0.672 |
| EEDS | 0.575 | 0.522 | 0.803 | 0.773 | 0.517 | 0.444 |
| SLIC | 0.524 | 0.507 | 0.867 | 0.851 | 0.736 | 0683 |
| ERS | **0.59** | **0.53** | **0.95** | **0.90** | **0.86** | **0.74** |

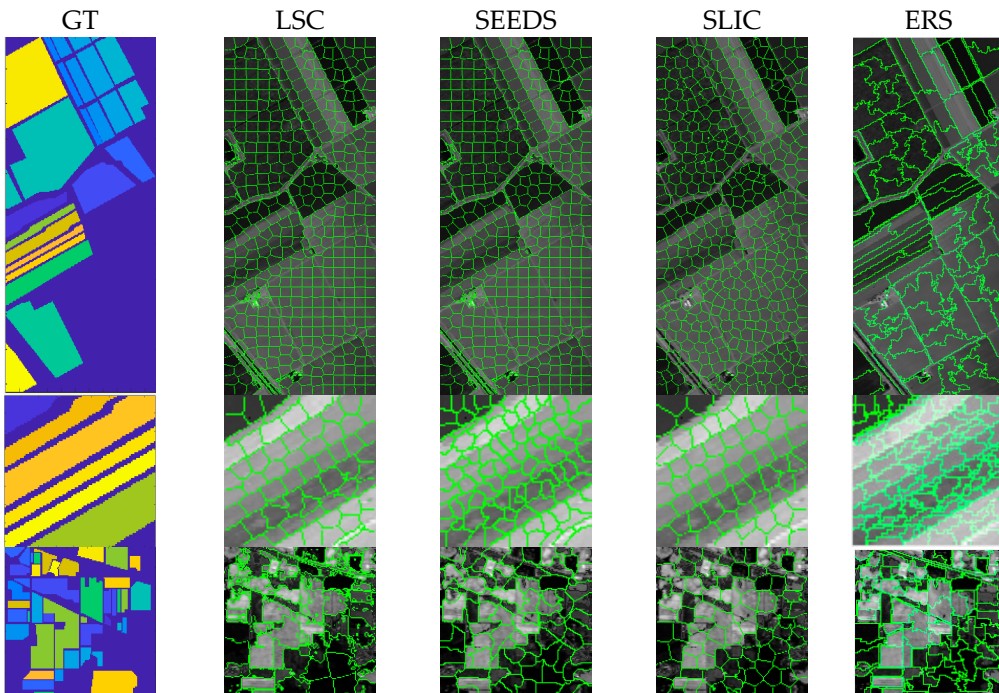

| GT | LSC | SEEDS | SLIC | ERS |

**Figure 6.** Segmentation results of different superpixel methods for SalinasA, Salinas and Indian Pines datasets (from top to bottom).

### 5.2. Experimental Results on the Self-Collected Dataset

First of all, Figure 7 shows the procedural results by gradually adding the proposed strategies, i.e., the baseline (B), baseline and data optimization (B + DO), baseline and data determination (B + DD), and the full version of the proposed approach. For comprehensive comparison, we tested our approach on both the labeled and the unlabeled pixels of the HSI dataset, respectively.

As seen, for the baseline approach, there were apparently misclassified pixels in the colored blocks. After adding the DO or DD strategy, these blocks were classified much better. When both the DO and DD strategies were added, the classification results were best, where the noisy fragments were significantly reduced. Please note that due to different color codes used in the segmented blocks, the final classification map may appear not the same as the ground truth. Rather, it reflects the real scenarios, which actually demonstrate that the proposed unsupervised HSI classification approach can even recover details that are neglected in the ground truth. In addition, on testing unlabeled data, our method could still distinguish different classes, further illustrating its superiority.

For quantitative evaluation, Table 3 compares the final classification results of the proposed method and three other representative unsupervised classification methods, i.e., spectral clustering (SC) [22], Nyström extension clustering (NEC) [31] and anchor-based graph clustering (AGC) [30]. The numbers in bold represent the best results in each group, where our PSSA almost outperformed all others in all compared groups.

It is worth mentioning that the compared SC algorithm took more time to solve the eigenvalue decomposition of the Laplacian matrix, leading easily to out of memory without any outputted classification results for relatively large datasets, e.g., Salinas and SC-1 to SC-5. Instead, both NEC and AGC worked well on large HSI datasets. However, NEC did not seem robust enough, due to possibly the indefinite affinity matrix and the plural elements contained in the inverse matrix. In addition, AGC slightly outperformed the NEC, owing to the utilized anchor-based spectral clustering, which was also the key reason why AGC was selected as the baseline in our approach. In addition to comparing with unsupervised methods, we also compared with two typical supervised classification

approaches, i.e., KNN [14] and SVM [46,47]. Not surprisingly, unsupervised methods produced worse results than the supervised ones; however, the gap is decreasing.

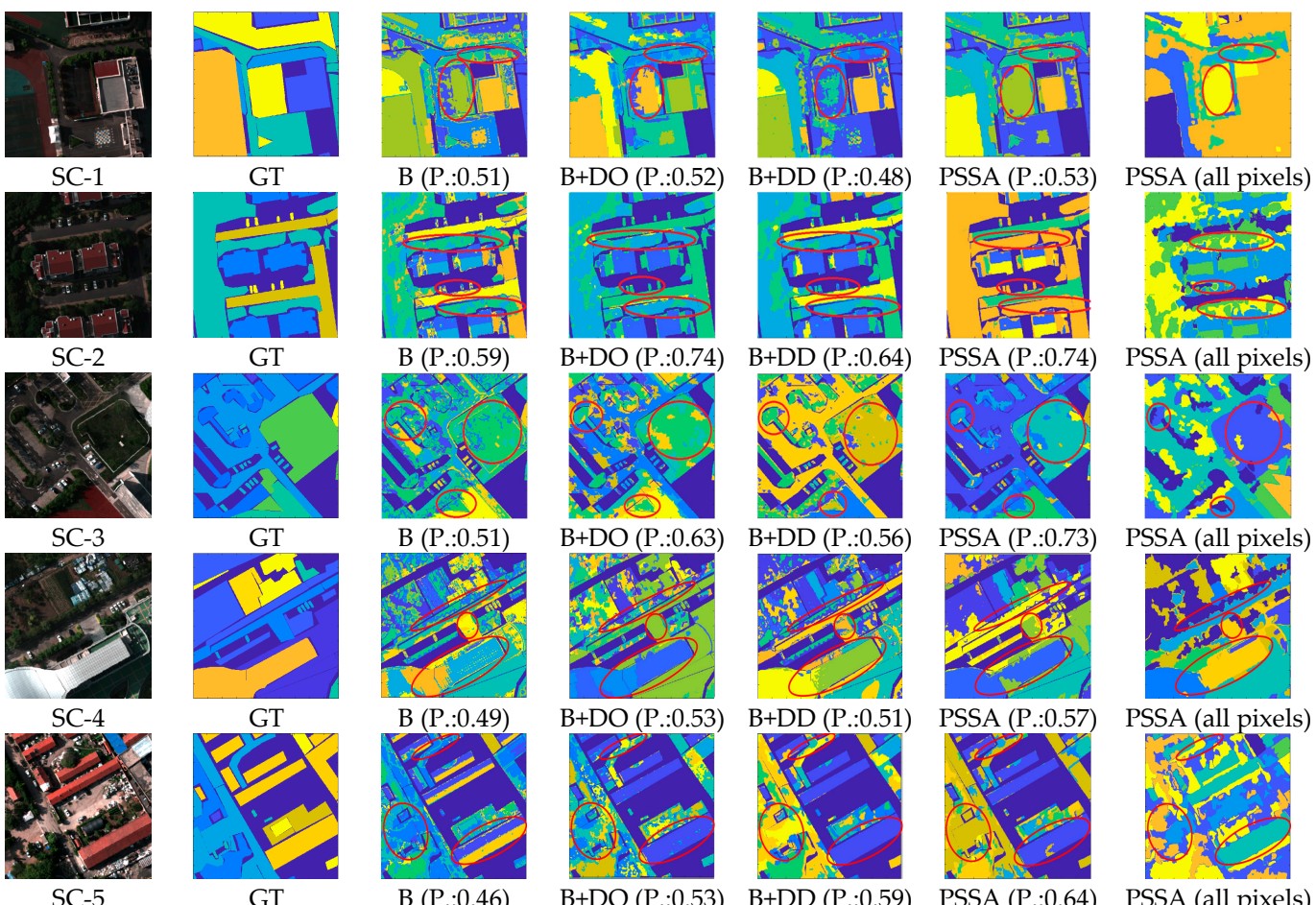

**Figure 7.** Original images of SC-1 to SC-5, their ground truths and the classification results in different steps. All the results are for labelled pixels only, except the last one from our approach. Areas in red circles in each row are for particular attention to compare the results of classification.

In detail, all the classification results were evaluated using five metrics, i.e., OA, AA, Kappa, P. and NMI. As seen in Table 3, the classification results of the proposed approach were significantly better than the other unsupervised ones. With the introduced two strategies in different steps, the classification performance gradually improved to be much closer to those from supervised classification algorithms.

## 5.3. Experimental Results on Publicly Available Datasets

In addition to the comparison using our own datasets, we also tested our approach on three publicly available HSI datasets, i.e., Salinas, SalinasA, and Indian Pines, with the experimental results shown in Figure 8. The overall trend of the classification results remains highly consistent with the findings from our own datasets, as discussed in detail below.

In Figure 8, compared with the baseline method, our algorithm improved the P. by 0.18 on Salinas, 0.11 on SalinasA, and 0.11 on Indian Pines. For the datasets with a relatively simple spatial distribution such as SaliansA and India Pines, our approach achieved a much higher improvement. However, for datasets with rather a more complicated spatial distribution, e.g., Salinas, the improvement achieved was relatively small. In addition, in order to ensure the integrity of the experiments, we verified the results on the whole image rather than on the labeled pixels alone.

**Table 3.** Clustering results on five self-collected datasets, where "NaN" represents no available results due to out of memory, and "-" denotes no referred results. Bold represents the best result of all methods and underline represents the best result of unsupervised methods.

| Datasets | Metrics | Unsupervised | | | | Supervised | |
|---|---|---|---|---|---|---|---|
| | | SC | NEC | AGC | PSSA | KNN | SVM |
| SC-1 | OA | NaN | 0.27 | 0.36 | 0.51 | 0.82 | **0.87** |
| | AA | NaN | 0.17 | 0.31 | 0.49 | 0.57 | **0.88** |
| | Kappa | NaN | 0.10 | 0.21 | 0.39 | 0.77 | **0.83** |
| | NMI | NaN | 0.27 | 0.28 | 0.38 | - | **-** |
| | P. | NaN | 0.46 | 0.51 | 0.53 | - | **-** |
| SC-2 | OA | NaN | 0.57 | 0.56 | 0.82 | 0.91 | **0.94** |
| | AA | NaN | 0.46 | 0.43 | 0.72 | 0.58 | **0.86** |
| | Kappa | NaN | 0.38 | 0.36 | 0.70 | 0.86 | **0.90** |
| | NMI | NaN | 0.40 | 0.38 | 0.50 | - | **-** |
| | P. | NaN | 0.72 | 0.59 | 0.74 | - | **-** |
| SC-3 | OA | NaN | 0.46 | 0.43 | 0.77 | 0.80 | **0.81** |
| | AA | NaN | 0.31 | 0.32 | 0.61 | 0.53 | **0.84** |
| | Kappa | NaN | 0.33 | 0.26 | 0.67 | 0.76 | **0.82** |
| | NMI | NaN | 0.35 | 0.43 | 0.52 | - | **-** |
| | P. | NaN | 0.44 | 0.51 | 0.73 | - | **-** |
| SC-4 | OA | NaN | 0.40 | 0.43 | 0.64 | 0.79 | **0.82** |
| | AA | NaN | 0.28 | 0.30 | 0.48 | 0.50 | **0.74** |
| | Kappa | NaN | 0.27 | 0.33 | 0.57 | 0.67 | **0.76** |
| | NMI | NaN | 0.37 | 0.41 | 0.56 | - | **-** |
| | P. | NaN | 0.43 | 0.49 | 0.57 | - | **-** |
| SC-5 | OA | NaN | 0.36 | 0.40 | 0.57 | 0.80 | **0.87** |
| | AA | NaN | 0.15 | 0.35 | 0.48 | 0.49 | **0.80** |
| | Kappa | NaN | 0.10 | 0.26 | 0.44 | 0.72 | **0.81** |
| | NMI | NaN | 0.22 | 0.34 | 0.44 | - | **-** |
| | P. | NaN | 0.38 | 0.46 | 0.64 | - | **-** |

Table 4 also shows quantitatively the classification results of the publicly available datasets. The NMI, P. and others all improved by about 0.1 to 0.2 after adding the DD or DO strategy, compared to the baseline. When both the strategies were used, the performance was further improved, again validating the efficacy of the proposed strategies.

By combining different strategies in the proposed approach, the classification results gradually improved, consistent with the results obtained from our own datasets in the previous subsection. The final results progressively approach that of the supervised classification using the KNN and SVM (both with 1% samples for training). For the unsupervised classification approach, NMI and P. are the only quantitative metrics, thus no results are given for SVM. It is worth noting that the supervised method can refer to prior knowledge for model training, so it can easily achieve a better classification effect. However, the unsupervised classification is closer to the actual application needs and is more difficult due to the lack of any prior reference information, which is the major difference between the unsupervised method and the supervised ones. Compared with supervised classification, unsupervised classification is a somewhat rough classification. In the case where OA

was less than 50%, our unsupervised classification approach could still perform a rough distinguishing classification. Whether to completely rely on the results of unsupervised methods will mainly depend on specific applications, especially the availability of labeled data as ground truth.

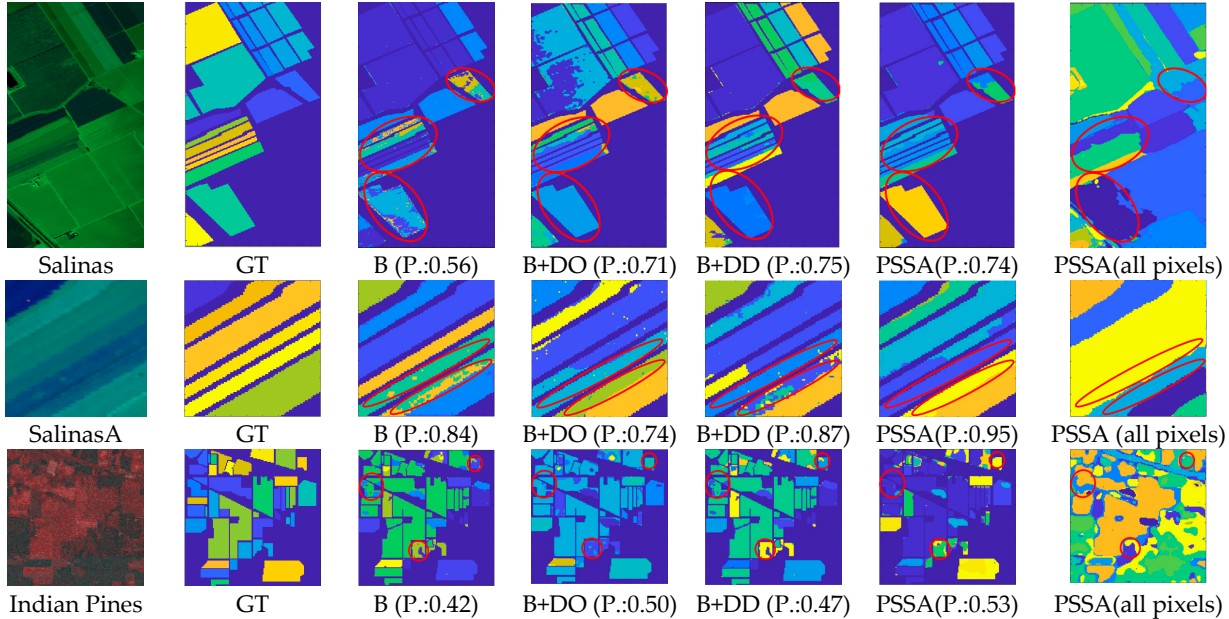

**Figure 8.** Original images of the Salinas (**top**), SalinasA (**middle**) and Indian Pines (**bottom**), the ground truths and the classification results. in different steps. All the results are for labelled pixels only except those from our approach. Areas in red circles in each row are for particular attention to compare the results of classification.

**Table 4.** Clustering results on three public datasets, where "NaN" represents unavailable results due to out of memory, and "-" denotes no referred results. Bold represents the best result of all methods, and underline represents the best result of unsupervised methods.

| Datasets | Metrics | Unsupervised | | | | Supervised | |
|---|---|---|---|---|---|---|---|
| | | SC | NEC | AGC | PSSA | KNN | SVM |
| | OA | NaN | 0.51 | 0.46 | <u>0.71</u> | 0.82 | **0.92** |
| | AA | NaN | 0.46 | 0.42 | <u>0.63</u> | 0.84 | **0.96** |
| Salinas | Kappa | NaN | 0.46 | 0.44 | <u>0.67</u> | 0.80 | **0.91** |
| | NMI | NaN | 0.72 | 0.71 | <u>0.86</u> | - | **-** |
| | P. | NaN | 0.60 | 0.56 | <u>0.74</u> | - | **-** |
| | OA | 0.74 | 0.72 | 0.65 | <u>0.82</u> | 0.94 | **0.98** |
| | AA | 0.72 | 0.69 | 0.61 | <u>0.84</u> | 0.81 | **0.98** |
| SalinasA | Kappa | 0.68 | 0.65 | 0.56 | <u>0.77</u> | 0.94 | **0.98** |
| | NMI | 0.80 | 0.77 | 0.81 | <u>0.90</u> | - | **-** |
| | P. | 0.81 | 0.78 | 0.84 | <u>0.95</u> | - | **-** |
| | OA | 0.40 | 0.37 | 0.42 | <u>0.49</u> | 0.52 | **0.73** |
| | AA | <u>0.41</u> | 0.32 | 0.23 | 0.33 | 0.44 | **0.62** |
| Indian Pines | Kappa | 0.32 | 0.29 | 0.30 | <u>0.38</u> | 0.44 | **0.69** |
| | NMI | 0.44 | 0.45 | 0.46 | <u>0.59</u> | - | **-** |
| | P. | 0.36 | 0.43 | 0.42 | <u>0.53</u> | - | **-** |

## 6. Conclusions

In this paper, a novel unsupervised hyperspectral classification approach, PCA-domain superpixelwise singular spectral analysis (PSSA), was proposed. Specifically, a data decontamination strategy, based on superpixel segmentation to generate homogeneous regions, was proposed to eliminate spatial redundancy, noise and intra-class inconsistency. In addition, by applying 2D-SSA in the PCA domain on the dimension reduced hypercube, a data optimization strategy was proposed to further remove noise and enhance features. Finally, the efficacy of the two proposed strategies was validated using the representative unsupervised classification framework, i.e., anchor-based graph clustering (AGC). As a result, the proposed method improved the classification accuracy by about 15–20% compared to several state-of-the-art methods. These strategies can be also introduced into other classification approaches. Experiments on several aerial remote sensing datasets have fully validated the efficiency and efficacy of the proposed methodology.

**Author Contributions:** Conceptualization, Q.L. and J.R.; methodology, Q.L. and D.X.; software, Q.L.; validation, Y.T.; formal analysis, J.R. and H.S.; investigation, Q.L. and D.X.; resources, Y.Z.; data curation, Y.Z.; writing—original draft preparation Q.L.; writing—review and editing, Q.L. and J.R.; visualization, Q.L. and J.R.; supervision, Q.L. D.X and J.R.; project administration, D.X.; funding acquisition, H.S. All authors have read and agreed to the published version of the manuscript.

**Funding:** This work is partially supported by the Key Laboratory of Airborne Optical Imaging and Measurement, Chinese Academy of Sciences (CAS), the International Cooperation Project of Changchun Institute of Optics, Fine Mechanics and Physics (CIOMP) (Y9U933T190), and the Dazhi Scholarship of the Guangdong Polytechnic Normal University.

**Data Availability Statement:** https://zyx980824.github.io/HSI-dataset/ (accessed on 2 June 2021).

**Conflicts of Interest:** The authors declare no conflict of interest.

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
