# Peer review of "PSSA: PCA-Domain Superpixelwise Singular Spectral Analysis for Unsupervised Hyperspectral Image Classification"

_remotesensing, doi:10.3390/rs15040890_

Round 1

Reviewer 1 Report

Please find my attached report.

Reviewer 2 Report

1. The name of Section 2 should be Related Works.

2. The components of the proposed method can be found in Section 2. That is to say, the proposed method simply combined existing methods. Hence, I think the main contribution of this manuscript is the collected dataset. The authors should re-consider the contributions.

3. I cannot find implement details. For example, how do the authors set the hyper-parameters? what's the configuration of used computer?

4. A lot of unsupervised methods for HSI Classification have been proposed like [1] Self-supervised learning with adaptive distillation for hyperspectral image classification; [2] SC-EADNet: a self-supervised contrastive efficient asymmetric dilated network for hyperspectral image classification. The authors should add more recent methods.

5. What's the number of training samples? The authors shoud add experiments on the effect of the number of training samples.

Reviewer 3 Report

The manuscript presents (remotesensing-2145528) a very interesting approach combining multivariate analysis to classification based unsupervised hyperspectral classification. All sections was well written and demonstrate relevant results to readers, as well as in Remote Sensing journal.

My only comment for this research is that the background introduction for economically applied by in relation an other analysis based multispectral and hyperspectral remote sensing techniques. It would be better if this background information could be provided.

- You should emphasize more on the novelty of the present methodology in abstract and conclusion as well.

#1: Please. All standardization of nomenclature equipment/reagents/software was performed when necessary. Example: Fabricant, City, State, Country (three-letter). Check all manuscript.

#2. Please check for “Author Instruction” and standardization to manuscript,

 #3. Alphabetic order keywords;

L.302. double dots?

Please, check Figures 3 and 4. Add statistical analysis and informer a sample number; In addition, quality is need to improve to readers.

Change Fig. To Figure or Figures;

Best regards,

Round 2

Reviewer 2 Report

The authors have answered all the concerns.